# Ischemic Post-Conditioning in a Rat Model of Asphyxial Cardiac Arrest

**DOI:** 10.3390/cells13121047

**Published:** 2024-06-17

**Authors:** Matthew B. Barajas, Takuro Oyama, Masakazu Shiota, Zhu Li, Maximillian Zaum, Ilija Zecevic, Matthias L. Riess

**Affiliations:** 1Department of Anesthesiology, Tennessee Valley Healthcare System, Veterans Affairs Medical Center, Nashville, TN 37212, USA; matthias.riess@vanderbilt.edu; 2Department of Anesthesiology, Vanderbilt University Medical Center, Nashville, TN 37212, USAzhu.li@vumc.org (Z.L.); max.zaum@googlemail.com (M.Z.); 3Department of Molecular Physiology and Biophysics, Vanderbilt University School of Medicine, Nashville, TN 37212, USA; shiotamasakazu@gmail.com; 4Department of Anesthesiology, University Medicine Greifswald, 17489 Greifswald, Germany; 5School of Medicine, Meharry Medical College, Nashville, TN 37212, USA; 6Department of Pharmacology, Vanderbilt University, Nashville, TN 37212, USA

**Keywords:** advanced cardiac life support, asphyxia, basic life support, compressions, cardiopulmonary resuscitation, ischemia-reperfusion injury, murine, resuscitation

## Abstract

Background: Ischemic post-conditioning (IPoC) has been shown to improve outcomes in limited pre-clinical models. As down-time is often unknown, this technique needs to be investigated over a range of scenarios. As this tool limits reperfusion injury, there may be limited benefit or even harm after short arrest and limited ischemia-reperfusion injury. Methods: Eighteen male Wistar rats underwent 7 min of asphyxial arrest. Animals randomized to IPoC received a 20 s pause followed by 20 s of compressions, repeated four times, initiated 40 s into cardiopulmonary resuscitation. If return of spontaneous circulation (ROSC) was achieved, epinephrine was titrated to mean arterial pressure (MAP) of 70 mmHg. Data were analyzed using *t*-test or Mann–Whitney test. Significance set at *p* ≤ 0.05. Results: The rate of ROSC was equivalent in both groups, 88%. There was no statistically significant difference in time to ROSC, epinephrine required post ROSC, carotid flow, or peak lactate at any timepoint. There was a significantly elevated MAP with IPoC, 90.7 mmHg (SD 13.9), as compared to standard CPR, 76.7 mmHg (8.5), 2 h after ROSC, *p* = 0.03. Conclusions: IPoC demonstrated no harm in a model of short arrest using a new arrest etiology for CPR based IPoC intervention in a rat model.

## 1. Introduction

Over 350,000 people are treated for out-of-hospital cardiac arrest (OHCA) yearly in the United States [1]. Disease burden remains high, despite advances in response systems and therapies delivered concomitantly to the implementation of cardiopulmonary resuscitation (CPR). Current CPR guidelines emphasize high-quality compressions with minimal interruptions [2]. While this serves to protect against further ischemic injury, it does not address injury from reperfusion. Abrupt restoration of blood flow after prolonged ischemic insult may result in molecular and metabolic changes more injurious than the ischemia itself [3,4]. Ischemic post-conditioning (IPoC) is a technique that has been developed to protect cells and tissues from the detrimental effects of ischemia-reperfusion (IR) injury (IRI) and its utility has been validated using in vitro, ex vivo, and in vivo models [5,6,7]. Researchers have found that the technique can be highly effective at reducing IRI in a number of different tissues and organs, including the heart, liver, kidney, and brain. Most notably, Yannopoulos and colleagues have used IPoC to drastically improve cardiac and neurologic functioning at 48 h after a prolonged 15 min ventricular fibrillatory arrest [8,9]. In their porcine study, they improved cerebral performance from coma (cerebral performance category score mean 3.8 out of 4, lower is better) to a mild to moderate deficit, mean 2.8, and significantly improved survival at 48 h. This demonstrated both the strength of this intervention and the large contribution reperfusion injury comprises in IRI.

Chest compression fraction, the measure of time in which chest compressions are being performed as compared to total arrest time, has been extensively studied and linked to survival [10]. Expert consensus suggests that a chest compression fraction of 80% or greater is ideal [11]. This is the basis for life support guidelines emphasizing minimization of chest compression interruptions surrounding automated external defibrillator use and pulse checks. Preventing further ischemia after reperfusion is undoubtedly important; however, alteration of an early reperfusion phase may be just as important. The timing of limited pauses early in reperfusion may be protective by reducing the production of reactive oxygen species, limiting mishandling of calcium and other ions, promoting the release of protective signaling molecules, and reducing inflammation [8,12].

Reperfusion injury is heterogenous and dependent upon the magnitude of the ischemic event. In fact, IRI is non-uniform. In the most extreme example, short bursts of IR without significant injury may even be protective [13]. Shorter ischemic insults may in fact have less reperfusion injury as a percentage of total injury. As IPoC targets reperfusion, in shorter ischemic arrests, its utility could be lessened or even detrimental [14]. This is because during pauses in compressions, coronary and cerebral perfusion pressures rapidly decrease. This fact may lead to the conclusion that IPoC should worsen outcomes, particularly when ischemic injury outweighs reperfusion injury. Often in non-witnessed arrests, the length of arrest prior to initiation of CPR is unknown. To have value clinically, IPoC must, at the least, do no harm to all patients. Thus, while demonstrated to be effective in long arrests, it is important to evaluate IPoC in shorter arrest periods as well. 

The young pig model is a high-fidelity model for human physiology; however, as IPoC would be an alteration to a life-saving procedure, complete and thorough investigation of this no-cost technique should occur. As such, re-demonstration of IPoC efficacy in a separate species in vivo would help solidify IPoC’s road to translation. Additionally, the etiology of arrest is usually unknown at the start of CPR. While the majority of adult OHCAs are cardiac in origin, hypoxic arrests are the second highest etiology in both adults and children [15,16]. We hypothesize that IPoC does not add to ischemic injury and improves outcomes even after short arrests in a new species, rat, and type of arrest, asphyxial.

## 2. Materials and Methods

Institutional Animal Care and Use Committee approval was obtained for all study procedures (Protocol M1800029-02, 6 January 2022, Vanderbilt University Medical Center, Nashville, TN, USA). Subjects were cared for in accordance with the Vanderbilt University Medical Center Division of Animal Care guidelines. Protocols were completed and data reported in accordance with the Animal Research: Reporting of In Vivo Experiments guidelines.

Eighteen adult male Wistar rats, 9 per group, 538 ± 38 g, were utilized. On the day of the arrest experiment, rats were anesthetized initially with isoflurane 5% for 5 min. Rats were then weighed, shaved, and intubated with a 14 gauge (G) catheter (BD Insyte Autogard, Becton, Dickinson and Co., Ltd., Franklin Lakes, NJ, USA) using video laryngoscopy [17]. The catheter was then sutured to the cheek to prevent dislodgement. Mechanical ventilation occurred, with inhalation of 1.5% isoflurane, F_i_O_2_ 40%, 8 cc/kg tidal volumes. Rectal temperature was monitored with target temperature maintenance between 36.5 and 37.5 °C using a heating mat (T/Pump Professional, Stryker Corp., Kalamazoo, MI, USA). 

Vascular access was then established via surgical cutdown as previously described [18]. The right jugular vein was cannulated with polyethylene 25 tubing (Instech Laboratories Inc., Plymouth Meeting, PA, USA) connected to a 25G blunt needle (Air-Tite Products Co., Inc., Virginia Beach, VA, USA). The right femoral artery and vein were cannulated in a similar fashion. Arterial flow was non-invasively measured with a vascular flowprobe (1PRB flowprobe, T402 flowmeter, Transonic Systemic Inc., Ithica, NY, USA) around the right carotid artery. Pressure transductions were performed using TruWave pressure transducers (Edwards Lifesciences Corp, Irvine, CA, USA). Hemostasis was achieved using a handheld thermo-cautery device (HIT1, Bovie High-temp cautery kit, Symmetry Surgical Inc., Antioch, TN, USA). 

After catheterization, inspired oxygen was reduced to room air, F_i_O_2_ 21%, for 5 min before intravenous administration of rocuronium (3 mg/kg, Sagent Pharmaceuticals Inc., Buffalo Grove, Il, USA). Concomitant with administration of rocuronium, the isoflurane concentration was reduced to 0.5%. Asphyxial arrest was induced 1 min (min) after administration of rocuronium by cessation of mechanical ventilation in the paralyzed animal and allowed to occur for 7 min.

Next, cardiopulmonary resuscitation (CPR) began by reinitiating ventilation now with 100% oxygen and randomization into standard CPR (S-CPR) or IPoC CPR protocols. In the IPoC protocol, after an initial standard CPR period of 40 s, interrupted chest compression cycles began. These included a 20 s pause in chest compressions followed by 20 s of reinitiating compressions comprising 1 round for a total of 4 rounds. After the 4 rounds of IPoC, compressions were continued in accordance with the standard CPR protocol as previously described [18]. All compressions occurred using an automated chest compressor with an adjustable depth at a rate of 200 per min. Compression depth was adjusted to target a diastolic pressure of greater than 23 mmHg during CPR. Epinephrine bolus, 2 mcg, was given after the completion of IPoC protocol or the corresponding time point in S-CPR, and rebolused every subsequent 5 min. Maximal CPR time was set a priori at 30 min. Following the return of spontaneous circulation (ROSC), epinephrine drip was titrated to a goal mean arterial pressure (MAP) of 70 mmHg. Fifteen min after ROSC, isoflurane was resumed at 1% and F_i_O_2_ was reduced to 50% for the remainder of the experiment. Lactate was measured using arterial blood sample in point of care device (Nova Biomedical Corp., Waltham, MA, USA). Experimental protocol is outlined in Figure 1.

Hemodynamic and vital sign data were recorded using Powerlab Series 16/30 in LabChart (Version 8.1.13, AD Instruments North America Inc., Colorado Springs, Inc., USA). Baseline and post-ROSC timepoints were marked in LabChart and data exportation was automated to limit selection bias. Missing data were not imputed. Five values from lactate data are missing due to device malfunction; central venous pressure (CVP) is missing from one animal due to a catheter clot. 

Power analysis was based on variation in epinephrine delivered in previous studies: 80% difference in means with 30% variation, adjusted for a shorter insult while maintaining a potentially clinically significant difference. To detect a 30% difference in means assuming a 20% standard deviation, equal group sizes, a sample size of 8 is required to achieve a power of 80% at an alpha of 0.05 as determined by PS: Power and Sample Size Calculator (v 3.1.2). Data normality was analyzed using the Shapiro–Wilk test. Data were considered non-normally distributed if even one time point did not meet normality. Carotid flow, heart rate, and MAP met normality criteria, and therefore all data are represented as means and standard deviations (SDs) and were analyzed using unpaired Student’s *t* test. The remaining variables are represented as medians and interquartile ranges (25th–75th) and were analyzed using the Mann–Whitney test. Significance was set at *p* ≤ 0.05, two-tailed. Statistical analysis was performed using Prism 9.1.2 (GraphPad Software Inc., La Jolla, CA, USA).

## 3. Results

There were no baseline differences in any hemodynamic or vital sign parameter. The rate of ROSC was identical between groups; eight of nine animals achieved ROSC in each group. All animals that survived suffered severe bradycardic arrest or pulseless electrical activity. In the two animals that did not achieve ROSC, this electrical activity quickly became fine ventricular fibrillation and then asystole. Time to return of spontaneous circulation (ROSC) is a marker of overall ischemic time and insult. The duration of CPR or time to ROSC was not statistically different, S-CPR 193 s (141–255) as compared to 153 s (139–228) with IPoC, *p* = 0.63, Figure 2A. Delivery of epinephrine to maintain mean arterial pressure at a minimum of 70 mmHg demonstrates the recovery of the heart after arrest. Less epinephrine would indicate a better resuscitation or mitigation of damage. The quantity of inotropic support in the form of epinephrine drip to maintain a MAP of 70 mmHg was not significantly different, 19.2 mcg (8.7–75.8) in S-CPR v. 11 mcg (2.9–138.7) in IPoC, *p* = 0.75, Figure 2B. 

Lactate is used as a marker of overall level of ischemic insult. There were no differences in lactate measurement between groups at any time point, baseline, 15 min or 2 h post ROSC, *p* = 0.66, 0.44, 0.19 respectively, Figure 3. Baseline lactate was identical and low in both groups, 1.6 mmol L^−1^, consistent with an appropriate surgical preparation. Within group analysis demonstrated a statistically significant increase in lactate from baseline to peak, *p* = 0.001 and 0.02 in S-CPR and IPoC, respectively. There was no significant difference between the groups (standard cardiopulmonary resuscitation (S-CPR) 3.7 mmol L^−1^ (interquartile range (IQR) 2.7–9.4) vs. ischemic post-conditioning (IPoC) 3.3 mmol L^−1^ (IQR 2.4–5.4), *p* = 0.44). Median lactate in both groups was reduced from the peak at 2 h, reaching 2.3 mmol L^−1^ (1.7–3.6) in IPoC and 3.6 mmol L^−1^ (2.1–5.2) in S-CPR. This demonstrates the clearing of lactate and rescuable phenotypes. Ns = 8/8, 8/7 and 6/6 for S-CPR/IPoC at each time point, respectively.

MAP is the most commonly evaluated parameter as an indication of recovery post arrest. There was no significant difference between groups at baseline (standard cardiopulmonary resuscitation (S-CPR) 113 mmHg (standard deviation (SD) 14) vs. ischemic post-conditioning (IPoC) 119 mmHg (SD 27), *p* = 0.61). All MAPs were above 100 mmHg after ROSC in the IPoC group, 161 mmHg (SD 10), while S-CPR showed a trend to a slightly lower but not significantly different MAP, 149 mmHg (SD 35), *p* = 0.36. This difference became significant at 2 h after arrest, with IPoC subjects demonstrating higher blood pressure free from epinephrine therapy, 91 mmHg (SD 14), compared to S-CPR 77 mmHg (SD 9), *p* = 0.03. N = 8 in both groups at all times, as displayed in Figure 4.

Carotid blood flow is a marker of perfusion to the brain and a surrogate for neuroprotection. Carotid flow was nearly identical at baseline between groups, *p* = 0.87. Carotid flows did not differ at 15 min after ROSC, *p* = 0.74. Despite an increase in 2 h MAP in IPoC, there was a decrease in flow in both groups at 2 h after arrest, but the groups did not significantly differ from each other: the mean for ischemic post-conditioning (IPoC) was 9.7 mL min^−1^ (standard deviation (SD) 6.3), compared to the mean for standard cardiopulmonary resuscitation (S-CPR) of 6.99 mL min^−1^ (SD 4.1), *p* = 0.33. N = 8 in both groups at all times, as demonstrated in Figure 5.

CVP can be used as a marker of right heart failure, especially after an insult such as asphyxial cardiac arrest. Here, there were no significant changes across time points or between groups either before or after arrest. At baseline, the median CVP was 4.7 mmHg (interquartile range (IQR) 1.1–13.0) in the standard CPR group (S-CPR), as compared to the ischemic post-conditioning (IPoC) group, where the median was 7.0 mmHg (3.9–13.6), *p* = 0.61. CVP did not differ between the groups at either 15 min or 2 h time points, *p* = 0.87 and 0.99, respectively, Figure 6. N = 7 in the standard CPR group and 8 in the IpoC group at all times.

HR is a key component of cardiac output and an indicator of recovery post arrest. Here, we see a slight elevation in heart rate immediately after arrest, which decreases by 2 h. There were no significant differences between groups at any time point, *p* = 0.07, 0.29 and 0.50, respectively. N = 8 in both groups at all times (Figure 7).

## 4. Discussion

This asphyxial cardiac arrest model was successful in causing a mild injury as demonstrated by a modest lactate peak elevation and the need for hemodynamic support in the form of epinephrine after ROSC. This model is well established in mimicking the core components of human CPR [18]. This short insult could potentially be equated to a witnessed OHCA or even an in-hospital arrest with a rapid response team resuscitation. Therefore, this context differs highly from the prior evaluation of IPoC after prolonged arrest in vivo [9,19]. In this shortened arrest, reperfusion injury is certainly milder than that in prior experimentation. This reduction in ischemic insult, consequently, reduces the associated reperfusion injury as well. As limiting reperfusion injury is the proposed mechanism of action of IPoC, in settings of limited reperfusion injury, IPoC may be of reduced effectiveness, or even be detrimental. This difference in ischemic insult from prior investigations evaluates whether IPoC remains beneficial or could potentially be detrimental by increasing ischemic injury when reperfusion injury is less severe. We found that there are no significant negative differences in any outcomes with potential improvement in post arrest MAP despite no difference in inotrope therapy. Thus, the major takeaway is that pauses initiated early in reperfusion do not worsen IRI even in short arrest scenarios. While our hypothesis was not proven correct, this finding is critically important to the potential clinical implementation of this powerful no-cost tool in the future. If IPoC was detrimental in the setting of short arrest periods, then a method to determine a length of arrest would be required. Thus, this study demonstrates that there is no barrier to the initiation of IPoC and that it is a viable strategy for further research in resuscitation of varied etiologies and lengths. 

Volatile anesthetics are known pre-conditioning agents, and their use in this study may have altered the response to IPoC. While mandatory for the humane utilization of the animal subjects in this study, this pre-conditioning effect was considered and, thus, the isoflurane dose reduced to the humane minimum, 0.5%, immediately before the initiation of arrest. The pre-conditioning effects of sevoflurane have been demonstrated to require, at a minimum, 1 MAC with a maximal effect reached at 1.5 or higher MAC [20]. Although we were under this delivered amount before CPR, some effect cannot be ruled out, but would have affected both groups. 

The gold standard marker for CPR success is recovery of neurological function. IPoC has been shown to improve neurologic outcomes after prolonged arrests; however, here, we predominantly demonstrated the cardiovascular protection of IPoC. While no difference in carotid blood flow at any point suggests no difference in ischemic insult to the brain, only neurologic testing in survival studies can definitively demonstrate this in the future. 

Here, we utilized a non-cardiac origin of arrest. Clinically, non-cardiac origin arrests account for approximately 20–40% of OHCAs and are increasing from causes such as COVID-19 and the opioid epidemic. Non-cardiac-origin arrest is associated with decreased rates of survival when compared to cardiac-origin OHCAs [15,21]. It is noted that it is unlikely in a clinical scenario to have an already established airway; however, there is no feasible way to hand-ventilate the animals, and intubation requires a temporary tilt to 45 degrees which would necessitate a large pause in compressions plus significant hemodynamic changes, compromising the study.

One key factor that demands future investigation is the role of co-morbidities in both the response to IRI and to post-conditioning. Advanced age and comorbidities such as diabetes mellitus are both prevalent amongst those who suffer OHCA and affect survival [22]. Age has been associated with alteration of the effects of ischemic conditioning, notably abrogating pre-conditioning while potentially remaining susceptible to post-conditioning [12]. Failure to account for these co-morbidities in pre-clinical work may lead to unexpected or unaccounted for variation in results when translation to clinical studies is attempted. 

## 5. Conclusions

Contradictory to current CPR guidelines, which emphasize the minimization of all pauses in compressions, short, repeated pauses timed early during reperfusion have repeatedly demonstrated benefit in pre-clinical models in mitigating organ injury after cardiac arrest. This investigation demonstrates the safety of this intervention after even short arrests, where ischemic injury may outweigh reperfusion injury. This conclusion additionally validates previous work in pigs undergoing prolonged cardiac arrest by ventricular fibrillation in a new species and etiology of arrest. IPoC is a no-cost, easily instituted technique that deserves further investigation.

## Figures and Tables

**Figure 1 cells-13-01047-f001:**
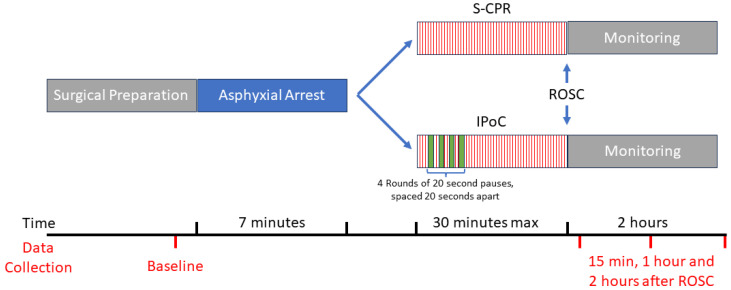
Experimental protocol. Details of flow and timing of experiments including 7 min arrest, cardiopulmonary resuscitation timing, and monitoring intervals.

**Figure 2 cells-13-01047-f002:**
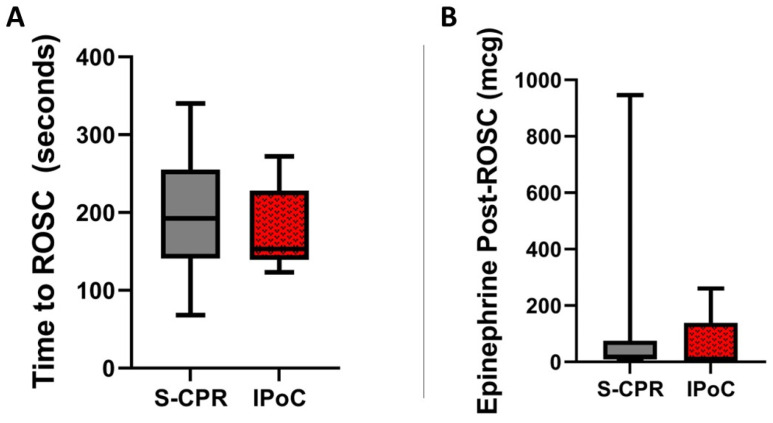
General markers of resuscitation quality. (**A**) Ischemic post-conditioning (IPoC) did not change the time to ROSC as compared to standard cardiopulmonary resuscitation (S-CPR). N = 8 in each group. (**B**). There was no statistically significant difference in epinephrine delivered, 19.2 mcg (interquartile range (IQR) 8.7–75.8) in S-CPR v. 11 mcg (IQR 2.9–138.7) in IPoC, *p* = 0.75. N = 8 in each group. In both figures, the box spans 25th to 75th percentile, the line in the box represents the median, and the whiskers extend from the 10th to 90th percentile.

**Figure 3 cells-13-01047-f003:**
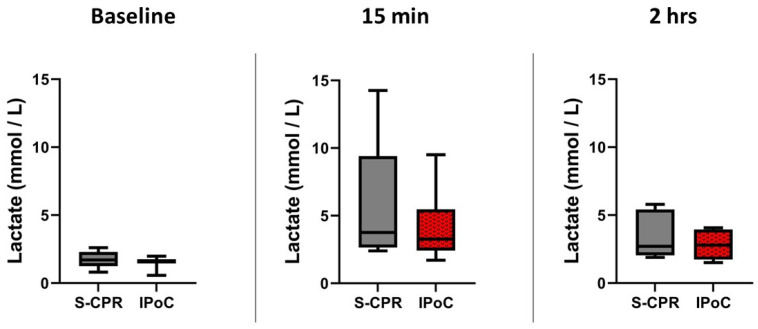
Lactate measurement before and after arrest. Lactate measurement is demonstrated at three time points: baseline, 15 min after return of spontaneous circulation (ROSC), and 2 h after ROSC, with no statistically significant difference between IPoC and S-CPR at any timepoint. In all figures, the box spans 25th to 75th percentile, the line in the box represents the median, and the whiskers extend from the 10th to 90th percentile.

**Figure 4 cells-13-01047-f004:**
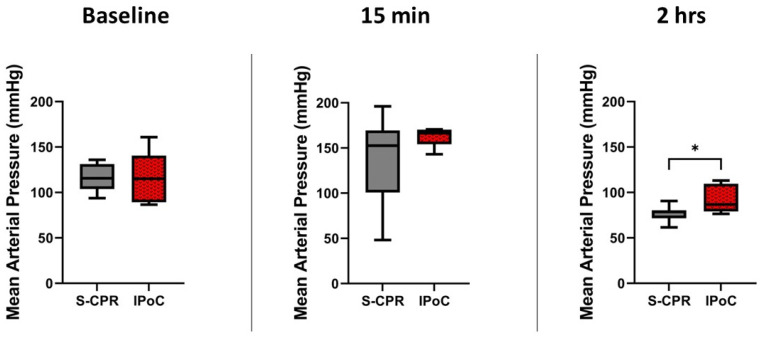
Mean arterial pressure before and after arrest. Mean arterial pressure (MAP) is demonstrated at three time points: baseline, 15 min after return of spontaneous circulation (ROSC), and 2 h after ROSC. MAP did not differ at baseline or 15 min post ROSC, *p* = 0.61 and 0.36, respectively. MAP at 2 h was significantly higher in IPoC, 91 mmHg (SD 14), compared to S-CPR, 77 mmHg (SD 9), * *p* = 0.03. In all figures, the box spans the 25th to the 75th percentile, the line in the box represents the median, and the whiskers extend from the 10th to the 90th percentile.

**Figure 5 cells-13-01047-f005:**
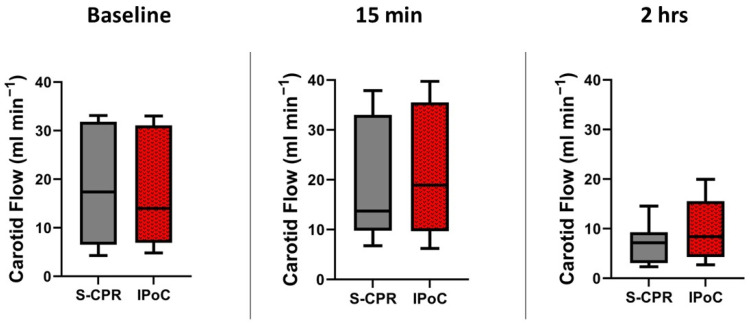
Carotid blood flow before and after arrest. Carotid blood flow is demonstrated at three time points: baseline, 15 min after return of spontaneous circulation (ROSC), and 2 h after ROSC. There was no significant difference in baseline carotid flow at any timepoint between groups. In all figures, the box spans the 25th to the 75th percentile, the line in the box represents the median, and the whiskers extend from the 10th to the 90th percentile.

**Figure 6 cells-13-01047-f006:**
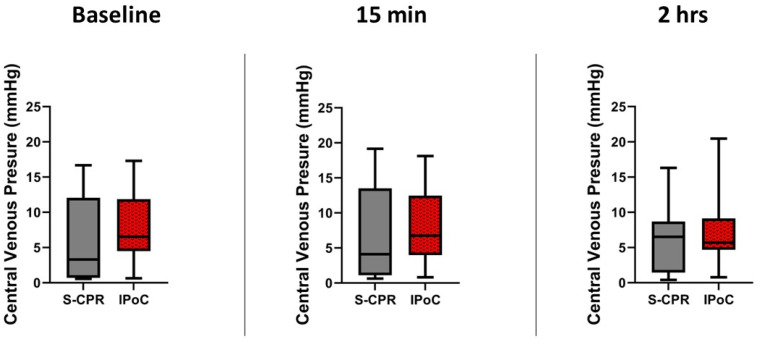
Central venous pressure before and after arrest. Central venous pressure (CVP) is demonstrated at three time points: baseline, 15 min after return of spontaneous circulation (ROSC), and 2 h after ROSC. There were no significant differences between groups at any timepoint. In all figures, the box spans the 25th to the 75th percentile, the line in the box represents the median, and the whiskers extend from the 10th to the 90th percentile.

**Figure 7 cells-13-01047-f007:**
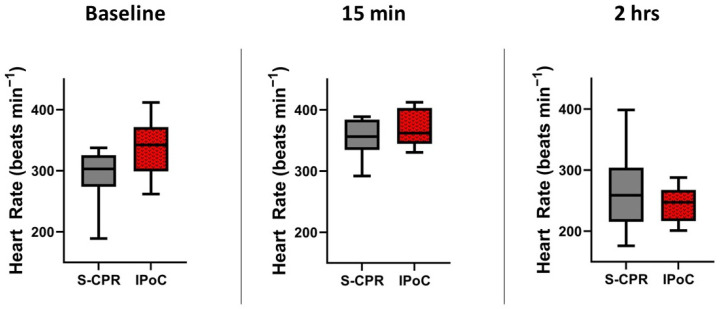
Heart rate. Heart rate (HR) is demonstrated at three time points: baseline, 15 min after return of spontaneous circulation (ROSC), and 2 h after ROSC. Heart rate did not differ between the groups at any time point including baseline, 15 min, or 2 h, *p* = 0.07, 0.29, and 0.50, respectively. In all figures, the box spans the 25th to the 75th percentile, the line in the box represents the median, and the whiskers extend from the 10th to the 90th percentile.

## Data Availability

All data are available upon reasonable request and in accordance with funding guidelines.

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
