# Peer review of "Ischemic Post-Conditioning in a Rat Model of Asphyxial Cardiac Arrest"

_cells, 2024, doi:10.3390/cells13121047_

Round 1
Reviewer 1 Report
Comments and Suggestions for Authors
This work aimed to assess the safety and efficacy of ischemic postconditioning (IPoC) in a model of asphyxial cardiac arrest. IPoC appears to be a promising procedure for reducing ischemia-reperfusion heart injury. However, IPoC, which was discovered about 20 years ago, has rarely been used in clinical practice due to ambiguous results of clinical trials, lack of standardization, and the need for further search for optimal protocols and investigation of clinical outcomes. The study of cardioprotective interventions on a non-cardiac model of cardiac arrest, used in this study, represents particular interest.
Comments:
Lines 25-26: “Conclusions: IPoC demonstrated no harm in a model of short arrest in a novel species with a new arrest etiology”. Why did the authors say, “novel species?” Rats have long been a popular choice for heart research.
Lines 27-28: Keywords should not be abbreviated.
Methods:
Lines 107-108: “Induction of asphyxial arrest was induced 1 minute (min) after administration of rocuronium and allowed to occur for 7 min”. Please explain how the asphyxia was achieved.
Please refer to the publication of Lamoureux L, Radhakrishnan J, and Gazmuri RJ. J Vis Exp, 2015 (Ref. 18) while explaining the experimental setup.
How was the 7 minutes of asphyxia chosen for the experimental protocol?
The study design of this work lacks markers of reperfusion injury (e.g., troponin I levels in the blood, markers of apoptosis, inflammation, or oxidative stress). Analysis of ventricular arrhythmias post-asphyxia would be appreciated. No biochemical parameters of pro-survival signaling were measured.
Lines 237-238: “This asphyxial cardiac arrest model was successful in causing a mild injury as demonstrated by a modest lactate peak elevation”. The elevation of lactate levels was statistically insignificant in both groups. One cannot report the change in the parameter if it is not statistically significant.
Author Response
REVIEWER#1
Lines 25-26: “Conclusions: IPoC demonstrated no harm in a model of short arrest in a novel species with a new arrest etiology”. Why did the authors say, “novel species?” Rats have long been a popular choice for heart research.
Thank you. The intent was that this was a separation from the prior work done by Yannopoulous and his team, which focused on pig models. To our knowledge this was the first attempt at CPR based IPoC after cardiac arrest. There has been a large amount of work focused on regional ischemia and coronary artery occlusion/reperfusion based IPoC. However, we have adjusted the wording based on your suggestion.
Lines 27-28: Keywords should not be abbreviated.
Thank you. While it was thought that common abbreviations may yield more searches as keywords, this has been changed at your suggestion.
Methods:
Lines 107-108: “Induction of asphyxial arrest was induced 1 minute (min) after administration of rocuronium and allowed to occur for 7 min”. Please explain how the asphyxia was achieved.
This has been clarified in the text. “…by cessation of mechanical ventilation in the paralyzed animal…”
Please refer to the publication of Lamoureux L, Radhakrishnan J, and Gazmuri RJ. J Vis Exp, 2015 (Ref. 18) while explaining the experimental setup.
This reference has been added. Lines 95-96, 116
How was the 7 minutes of asphyxia chosen for the experimental protocol?
Given previous work by other authors demonstrating the utility of IPoC in CPR after very prolonged arrest times, we wanted to ensure IPoC safety when reperfusion injury was minimal. The arrest time was titrated downward in successive experiments while ensuring incomplete rescue, <100% ROSC yet a significant lactate bump indicating some damage occurred.
The study design of this work lacks markers of reperfusion injury (e.g., troponin I levels in the blood, markers of apoptosis, inflammation, or oxidative stress). Analysis of ventricular arrhythmias post-asphyxia would be appreciated. No biochemical parameters of pro-survival signaling were measured.
Thank you. We have added discussion of the rhythm status to the manuscript. As this was a mild injury model we have foregone additional injury markers, in future studies of at risk comorbidities, and more severe injury models we will absolutely explore these. Thank you for the input.
Lines 237-238: “This asphyxial cardiac arrest model was successful in causing a mild injury as demonstrated by a modest lactate peak elevation”. The elevation of lactate levels was statistically insignificant in both groups. One cannot report the change in the parameter if it is not statistically significant.
Thank you. We have clarified this point in the text. The lactate was not significantly different between groups at any timepoint. However, within group change from baseline to peak was significantly elevated in each group.
Reviewer 2 Report
Comments and Suggestions for Authors
The manuscript “Ischemic Post-Conditioning in a Rat Model of Asphyxial Cardiac Arrest” by Matthew B. Barajas et al. used an asphyxial cardiac arrest model to compare standard CPR (S-CPR) vs Ischemic post-conditioning (IPoC) CPR protocols. The authors found that IPoC does not worsen outcomes in their model.
In the introduction, the authors hypothesize that IPoC does not add to ischemic injury and improves outcomes even after short arrests in their rat model of asphyxia cardiac arrest. They demonstrate that IPoC did not change the time to ROSC and that lactate levels, a marker of the overall level of ischemic insult, were compared to S-CPR. Yet, IPoC failed to improve outcomes, which they barely addressed in the discussion. Could the authors further speculate why their hypothesis failed? In additional studies, how could they demonstrate that IPoC is protective in their model?
The manuscript would benefit from adding another figure with a diagram comparing the interventions (S-CPR vs. IPoC) and the times of the measurements before and after arrest.
In the result section (lines 181-189), the authors address all the data shown in Figures 3-6 in a very general manner. Yet, the figure legends for Figures 3-6 are highly detailed and explain the results in more depth. It should be the other way around. Also, each figure should appear after the text that mentions it. Here, a paragraph describes all three figures (3-6), and then all figures appear together with no text between them.
Author Response
REVIEWER#2
The manuscript “Ischemic Post-Conditioning in a Rat Model of Asphyxial Cardiac Arrest” by Matthew B. Barajas et al. used an asphyxial cardiac arrest model to compare standard CPR (S-CPR) vs Ischemic post-conditioning (IPoC) CPR protocols. The authors found that IPoC does not worsen outcomes in their model.
In the introduction, the authors hypothesize that IPoC does not add to ischemic injury and improves outcomes even after short arrests in their rat model of asphyxia cardiac arrest. They demonstrate that IPoC did not change the time to ROSC and that lactate levels, a marker of the overall level of ischemic insult, were compared to S-CPR. Yet, IPoC failed to improve outcomes, which they barely addressed in the discussion. Could the authors further speculate why their hypothesis failed? In additional studies, how could they demonstrate that IPoC is protective in their model?
You are correct that our hypothesis was not proven correct. We believe this is due to the relative ratio of ischemic to reperfusion injuries after mild insult. The longer the ischemic insult the more relative weight the reperfusion injury holds compared to the ischemic injury. With this insult we did not reach a relative ratio where limiting reperfusion injury at the expense of increased, albeit short, ischemic injury was beneficial. Importantly, neither was it detrimental to the animal. Increasing the ischemic injury, by lengthening ischemic time or by adding co-morbidities would increase relative weight of the reperfusion injury thereby increasing the effectiveness of our intervention. Prolonged arrest IPoC has been performed before in other animal models to great effect. What has not been previously shown is how effective or safe this intervention is after short arrests, and if it is worth pursing in humans where ischemic length is often unknown.
We have expanded our address of this fact in the first paragraph of the Discussion section.
The manuscript would benefit from adding another figure with a diagram comparing the interventions (S-CPR vs. IPoC) and the times of the measurements before and after arrest.
This is an excellent suggestion and has been added as figure 1.
In the result section (lines 181-189), the authors address all the data shown in Figures 3-6 in a very general manner. Yet, the figure legends for Figures 3-6 are highly detailed and explain the results in more depth. It should be the other way around. Also, each figure should appear after the text that mentions it. Here, a paragraph describes all three figures (3-6), and then all figures appear together with no text between them.
Thank you for this suggestion to improve readability. Each result and its associated figure is now isolated and with the other formatting suggestions followed.

Round 2
Reviewer 1 Report
Comments and Suggestions for Authors
In the revised version of the manuscripts, the authors have clarified the critical points that improved the manuscript quality.
Comments on the Quality of English LanguageSome minor grammar and stylistic errors are present in the text.